# Primary Teeth Supported Fixed Prosthesis—A Predictable Treatment Alternative

**DOI:** 10.3390/children9060804

**Published:** 2022-05-30

**Authors:** Sarit Naishlos, Liat Chaushu, Oded Ghelfan, Joseph Nissan, Benjamin Peretz, Tal Ratson, Gil Ben-Izhack, Moshe Davidovich, Sigalit Blumer

**Affiliations:** 1Department of Pediatric Dentistry, The Faculty of Medicine, The Maurice and Gabriela Goldschleger School of Dental Medicine, Tel Aviv University, Tel Aviv 69978, Israel; river554@gmail.com (S.N.); bperetz@tauex.tau.ac.il (B.P.); talrdmd@gmail.com (T.R.); blumer@012.net.il (S.B.); 2Department of Periodontology and Implant Dentistry, The Faculty of Medicine, The Maurice and Gabriela Goldschleger School of Dental Medicine, Tel Aviv University, Tel Aviv 69978, Israel; liat.natanel@gmail.com; 3Department of Oral Rehabilitation, The Faculty of Medicine, The Maurice and Gabriela Goldschleger School of Dental Medicine, Tel Aviv University, Tel Aviv 69978, Israel; drghelfanoded@gmail.com (O.G.); gil.ben.izhack@gmail.com (G.B.-I.); davidom@post.tau.ac.il (M.D.)

**Keywords:** anodontia, fixed restoration, tooth, deciduous, all-ceramic restoration

## Abstract

Background: Individuals with tooth agenesis often present a significant clinical challenge for dental practitioners. This retrospective study evaluated clinical and radiological long-term functional and esthetic outcomes following restoration using primary teeth to support fixed all-ceramic prosthesis in patients with teeth agenesis. Methods: Patients with teeth agenesis and at least one year follow-up were included. Examinations included panoramic X-ray, clinical examination and family history records. Only primary teeth without permanent teeth underneath were chosen. All ceramic fixed restorations were used. All data were collected from patient files. Outcome parameters included: restoration parameters (restoration survival, restoration fractures, restoration detachment, restoration replacement, and secondary caries), plaque index, and gingival index. Results: The study included 58 porcelain restorations inserted in 25 individuals; mean age 12 ± 2.1 years (range 10–19 years); mean number of missing teeth 12.3 ± 9 (range 6–12). Mean follow-up 48 ± 6 months (range 12–60 months). All restorations survived up to last follow-up, rendering a survival rate of 100%. Restorations outcome—porcelain chipping (9%), detachment (2%), no restoration replacement nor secondary caries, mean gingival index—0.7 ± 0.5 and mean plaque index—0.9 ± 0.3. Conclusions: In tooth agenesis, restoration using primary teeth to support fixed all-ceramic prosthesis is a viable treatment alternative.

## 1. Introduction

Anodontia is a prevalent malformation in humans. It may occur as a part of a syndromic manifestation or as a nonsyndromic isolated trait [1]. It can also be associated with oral clefts and several other syndromes [1,2]. Other conditions that have tooth agenesis as one of their features include Down’s syndrome and ectodermal dysplasia [3,4,5,6]. In these syndromes, there is a characteristic pattern of agenesis, usually different from the overall population [3,4,5,6].

The reported prevalence ranges from 2.2% to 15.68%, depending on the population studied, excluding third molars [3,7,8,9]. Most affected individuals lack only one or two teeth; permanent second premolars and upper lateral incisors are usually absent [1,8,9,10].

Individuals with tooth anodontia recurrently present an essential clinical problem to the dental practitioners due to the need for a multidisciplinary approach [3,4,5,10]. In many cases, this leads to prolonged treatment time and unfavorable outcomes, requiring re-evaluation at short intervals [3,4,5].

Restorations in young individuals presenting tooth anodontia are considered an essential pratice leading to improvements in their overall general feeling. Teeth absence might lead to chewing difficulties, dietary insufficiencies, communication complications, and unfavorable appearance. In such cases, prosthetic treatment is a significant major step to accomplish the expected functional, esthetic, and psychological aims. Compared to removable dentures, fixed restorations are relatively agreeable to patients and promote more constant superior hygienic and esthetic outcome. While, improving speech and masticatory function and there is good compliance by children, which makes fixed prostheses the ultimate solution to children suffering from anodontia [4,5].

Primary tooth supported restorative treatment planning presents an exceptional challenge [3] and should take into consideration alterations of tooth morphology [10,11,12,13,14,15,16]. Teeth deficiency may cause unfavorable restriction of bone structure, leading to an additional complication of implant supported restorations [17,18]. Moreover, the desire for a pleasing esthetic appearance is very high in young patients [4,5,16]. Consequently, young children presenting tooth agenesis require early treatment to overcome esthetic, functional (food intake, development of speech), and social challenges [10,11,12,13,14,15,16,17,18,19]. The insertion of implants for the reconstruction of implant-supported dentures is rarely an alternative at young age.

Immovable restorations for anodontia using primary teeth is relative novel and emerging treatment technique that can help to overcome many problems produced through removable appliances. It suitable for the growing process in the lower jaw, reduces the necessity to redo the removable dentures over time, and present improved visual result. Pediatric dentists are experiencing increased demand from parents to enhance esthetics when treating young patients’ teeth.

Therefore, esthetic prefabricated fixed restorations for deciduous frontal teeth are becoming more widely used by pediatric dentists for restorations in young patients suffering from anodontia.

Developments in the area of porcelain restorations chemically attached to enamel or dentin have led to a practical and predictable treatment alternative to reconstruct teeth, presenting a variation in form in young patients with increased esthetic demand, while using conservative techniques for preserving tooth structures [20,21,22].

A retrospective cohort study evaluated prefabricated zirconia crowns (ZC) used for deciduous restoring maxillary incisors; the treatment conditions were general anesthesia or sedation and previous tooth pulpotomy. The findings suggested that ZC achieved satisfactory clinical results, such as restorations’ adaptation and appearance using deciduous teeth [23]. The importance and uniqueness of the current study is the evaluation of the usefulness of zirconia crowns in restoring vital deciduous teeth.

The purpose of this retrospective study was to clinically and radiologically assess the long-term functional and esthetic outcomes following restoration of tooth agenesis using primary teeth to support fixed all-ceramic prosthesis. The study hypothesis was that primary teeth may serve as predictable abutments for tooth-supported fixed ceramic restorations.

## 2. Materials and Methods

Twenty-five patients were included in the study. They all attended the pedodontic and prosthodontic departments at the School of Dentistry at Tel Aviv University. The patients’ age was from 10–19 years old, according to treatment eligibility given by the Ministry of Health. All patients had a reduced number of permanent teeth (Figure 1).

The inclusion criteria were: tooth agenesis and at least one year follow-up post treatment. All patients were included after a meticulous evaluation of their medical histories and dental examinations, including panoramic X-ray (Figure 2), clinical examination and family history records. Only primary teeth without permanent teeth underneath were chosen as abutments to avoid lack of eruption. Permanent teeth were left untreated. All patients compatible with the inclusion criteria were included due to the relatively low numbers in the population.

All procedures were fully explained to the patients and parents and alternative treatment and/or material were offered. The study was approved Ethics Committee of Tel Aviv University.

The exclusion criteria were: patients under the age of eight years old, a lack of cooperation with dental treatment, and primary teeth mobility.

Restorative technique highlights–A minimal primary tooth preparation (eliminating undercuts) was advocated. In order to preserve maximum enamel and allow a direct path for prosthesis insertion including parallel tooth walls in order to achieve a retentive form restoration. A local anesthetic with 2% xylocaine and epinephrine 1:100,000 (lidocaine HCl and epinephrine injection, USP, DENTSPLY Pharmaceutical, Charlotte, NC, USA) was administered for all cases. A one-stage impression technique was performed using polyvinyl siloxane-based impression materials (3M ESPE express VPS impression material). Before cementation, the ceramic restorations were clinically verified for marginal adaptation, contour, and color match. The crown thickness ranged from 0.5 to 1.0 mm at the cervical, buccal, and palatal surfaces, whereas the incisal part was 1.5 mm thick. For the ideal esthetic outcome, all patients were rehabilitated with monolithic zirconia restorations (Prettau, Zirkonzahn, Gais, Israel). Resin-reinforced glass ionomer luting cement (Fuji Plus, GC Corp., Tokyo, Japan) was used for all tooth-supported restorations (Figure 3).

The clinical follow-up included clinical examination radiographs and clinical photographs, which revealed excellent marginal contour, and patients’ satisfaction with the restorations. Analysis of clinical and radiological parameters was performed every 6 months in the first year and then every year, and included restoration parameters (restoration survival, porcelain chipping, restoration detachment, restoration replacement, secondary caries), plaque index, and gingival index (0–3 according to Loe and Silness) [24]. All the data were collected from the patient files. Descriptive statistical analysis was used to describe the data collected.

## 3. Results

The study group comprised 58 porcelain restorations inserted into 25 individuals; the mean age was 12 ± 2.1 years (range 10–19 years) and the mean number of missing teeth was 12.3 ± 9 (range 6–12). The distribution of restorations (Table 1) was primary first and second molars (24/58—41.37%) and incisors and canines (34/58—58.63%). The mandible contained 34/58 (58.62%) of the restorations while 24/58 (41.38%) were placed in the maxilla.

The mean follow-up was 48 ± 6 months (range 12–60 months). All restorations survived up to last follow-up, rendering a survival rate of 100%.

Other restoration parameters were: porcelain chipping (3 prostheses, 9%), restoration detachment (2 prostheses, 6%). There was no restoration replacement nor secondary caries detection at the radiographic analysis (Figure 4 and Figure 5).

The gingival index was used for the assessment of prevalence and severity of gingivitis. Score 0 = Normal gingiva; Score 1 = Mild inflammation—slight change in color, slight edema, no bleeding on probing; Score 2 = Moderate inflammation—redness, edema, glazing, bleeding on probing; Score 3 = Severe inflammation—marked redness and edema, ulceration, tendency toward spontaneous bleeding. Gingival parameters included: gingival index—mean 0.7 ± 0.5, and plaque index—mean 0.9 ± 0.3.

No relationships were found between parameters.

## 4. Discussion

The prosthetic rehabilitation of patients with teeth anodontia needs cautious planning in order to produce restorations that suit their requests and have a lack a harmful consequence on their quality of life. Treatment options comprised removable partial dentures, overdentures, and fixed partial dentures (FPDs). Removable partial appliances are the most prevalent treatment alternative due to the relatively low cost, uncomplicated manufacturing and adjustment, but alternatively present complications such as denture retention and recurrent modifications. Usually, the denture-retentive quality is very limited because of the underdevelopment of alveolar ridges; consequently, removable partial dentures become disfavored. Overdentures have better retention comparing to removable partial dentures; however, they necessitate devitalization of the vital abutment teeth. Therefore, overdentures are not considered the preferred treatment alternative due to the unconservative preparations. Commonly, primary teeth receive composite restorations (direct or indirect fabrication) and maintenance is required over the years (color changing, composite fractures); hence, the approach of using new ceramic material in younger patients has become popular despite being initially more expensive. The ceramic FPDs treatment alternative present good properties for stability, retention, esthetics, and patient comfort. Consequently, FPDs have become commonly requested by parents and young patients for treatment of anodontia cases, due to good esthetics, color stability, and enhanced retention and restorations stability.

The wear of primary teeth is dependent on the enamel hardness, enamel, and dentin thicknesses, and biting forces in children, although tooth wear may occur also from different dental materials. A study evaluating the wear of deciduous tooth enamel using diverse restorative martials (monolithic zirconia, lithium disilicate glass ceramic, resin nanoceramic, and nanohybrid composite resin) indicated that zirconia produces reduced antagonist tooth wear compared to the other restorative materials used. Thus, it may be recommended for fabricating fixed restorations for deciduous teeth prostheses [25].

Fixed prostheses including rigid connectors are frequently avoided in actively growing patients since they might impact jaw growth, particularly if the prosthesis crosses the midline. Consistent with Barrow and White [26], intercanine width is determined at the ages of 5–8 years and created by distal movement of deciduous canines into the primate spaces to provide space for the erupting permanent incisors. The lack of lower permanent incisors causes early canine eruption, leading to little intercanine growth. The principal growth will be distal to the last deciduous tooth to accommodate the eruption of the permanent teeth, by which arch length is increased. Consequently, only patients ≥ 10 years old were included in the present study.

Currently, the use of dental implants in children is popular. However, the transverse growth of the maxilla continues up to the age of 17 in boys when the midpalatine suture fuses, which contraindicates the use of maxillary dental implants in young patients. Therefore, the placement of dental implants in a growing patient carries the risk of growth cessation, implant submergence, or ankylosis. Moreover, the placement of dental implants in patients with tooth agenesis is challenging due to deficiency in bone quantity and quality, in addition to the constant prosthesis modifications,

Cronin et al. [27] determined that implant rehabilitation should begin after 15 years of age for girls and after 18 years of age for boys to provide long-term prognosis with minimum complications.

The results of this study present the excellent clinical and radiological performance of ceramic-based restorations bonded to tooth structure in primary teeth in terms of marginal integrity, esthetics, and periodontal health over a mean follow-up period of 48 ± 6 months (range 12–60 months) (Figure 6, Figure 7, Figure 8, Figure 9 and Figure 10).

Recent clinical evidence concerning dental treatment options and treatment consequences in patients with tooth agenesis is still lacking [28]. The quality of studies is problematic; most studies focus on implant treatment and few have reported other treatment modalities [29].

Early prosthetic rehabilitation is important from functional, esthetic, and psychological perspectives. Congenital absence of teeth (oligodontia) also has an impact on a child’s emotional state and quality of life. Moreover, because treatment is usually time-consuming and exhausting, it may affect the child’s parents, relationship between the parents, and entire family [30].

Treatment alternatives of partial anodontia differ depending on the existing teeth, mainly on the root and the crown condition [31]. When the root and the crown are in good condition and esthetic improvement is required, the deciduous tooth can be reshaped, while ceramic crowns can be used for sound teeth with improper shape or size [32]. Although such a treatment is more costly compared to direct composite, its advantages (conservative tooth preparations, bonded to tooth structure, good esthetic, color, and restoration stability) make it an attractive treatment option.

Treatment with primary teeth-supported fixed prostheses is rarely described [28]. Our experience with patients with teeth agenesis indicates that a fixed tooth-supported prosthesis is promising due to favorable deciduous teeth distribution and the ability to cover their unfavorable shape (microdontia or taurodontia). The prosthetic treatment is conservative; only undercuts are removed, rendering almost complete tooth substance preservation. The results of the present study demonstrate that children adapt easily to fixed restorations; thus, they can be considered as a valuable treatment option. The null hypothesis was confirmed.

The gingival parameters in the present study were favorable, emphasizing the ease of maintaining oral hygiene despite the patients’ young age.

Periodic dental recall of patients with teeth agenesis must be performed at regular intervals in order to monitor the patient’s growth and adjust or replace the prosthesis accordingly. Oral hygiene should be maintained by using a fluoridated dentifrice twice daily; a microbrush or superfloss should be used to clean around the artificial teeth; and topical fluoride varnish should be applied in the dental clinic.

The findings of the present study suggest that ceramic-based restorations supported by primary teeth are an esthetic and biocompatible solution for missing teeth.

Previous studies also demonstrated marginal integrity, gingival health, and biocompatibility of zirconia crowns with little or no gingival inflammation adjacent to the restorations at up to 24 months of follow-up [23]. The ceramic-based prosthesis seemed to be resistant over time (mean follow-up 48 ± 6 months) with preservation of surface gloss with zirconia crowns on deciduous molars [33].

Oligodontia may cause functional and psychological impacts on the oral health-related quality of life (OHRQoL) [34,35,36,37]. Moreover, the impacts of oligodontia (oral symptoms, functional limitations, emotional wellbeing, social wellbeing) reported by children were significantly higher than those reported by their parents, indicating that children suffer more compared to what was perceived by their parents. There was a significant correlation between overall impacts reported by affected children and the number of site-specific tooth absences. This emphasizes the need for oral rehabilitation to minimize site-specific tooth absences [35].

## 5. Conclusions

Although there is a lack of evidence supporting treatment modalities of patients with tooth agenesis, restoration using primary teeth to support fixed all-ceramic prosthesis is a viable treatment alternative. Within the limitations of the present study (retrospective, one-center) in patients with tooth agenesis, further clinical and radiological studies are needed to determine the efficacy of ceramic prosthesis in restoring teeth using primary teeth as final abutments.

## Figures and Tables

**Figure 1 children-09-00804-f001:**
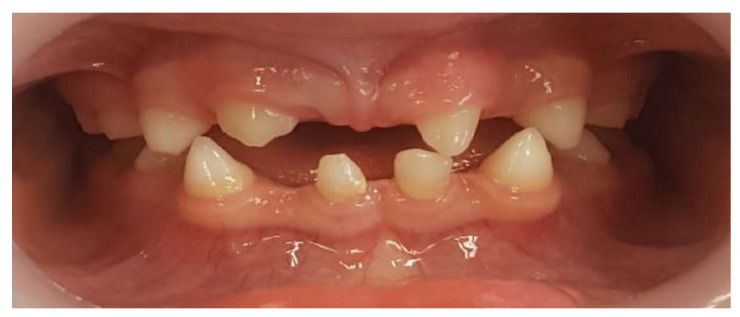
Ten-year-old boy presenting with tooth agenesis.

**Figure 2 children-09-00804-f002:**
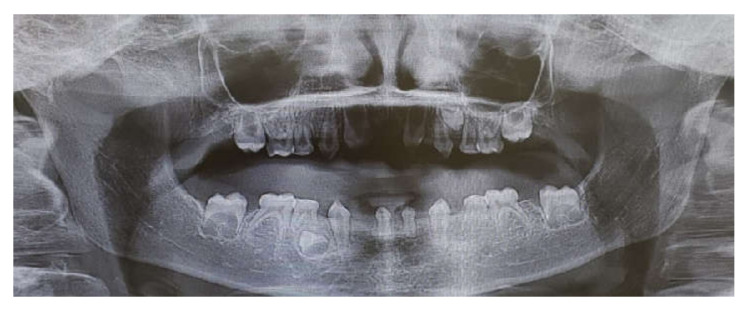
Panoramic pre-operative X-ray.

**Figure 3 children-09-00804-f003:**
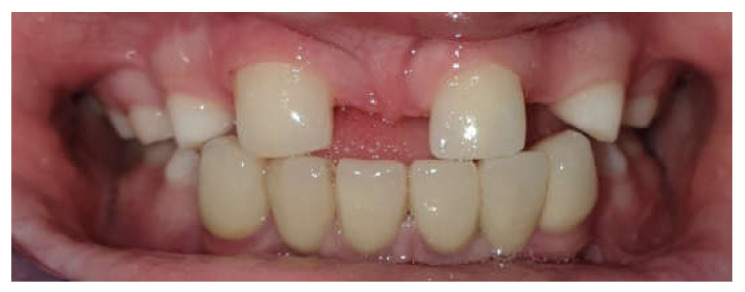
Porcelain teeth restoration 43–33. Post-operative, 36 months’ follow-up.

**Figure 4 children-09-00804-f004:**
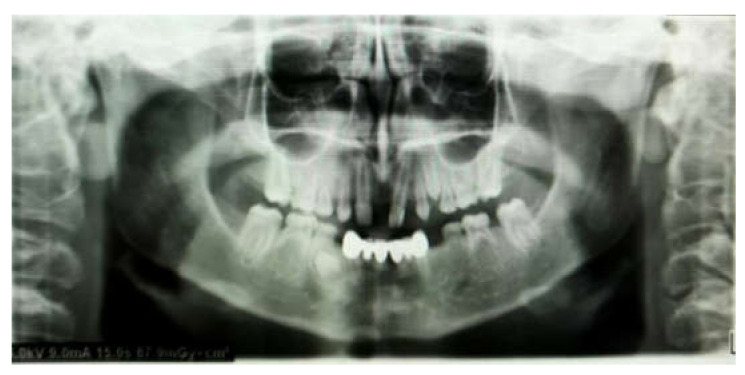
Panoramic X-ray. Post-operative, 36 months’ follow-up.

**Figure 5 children-09-00804-f005:**
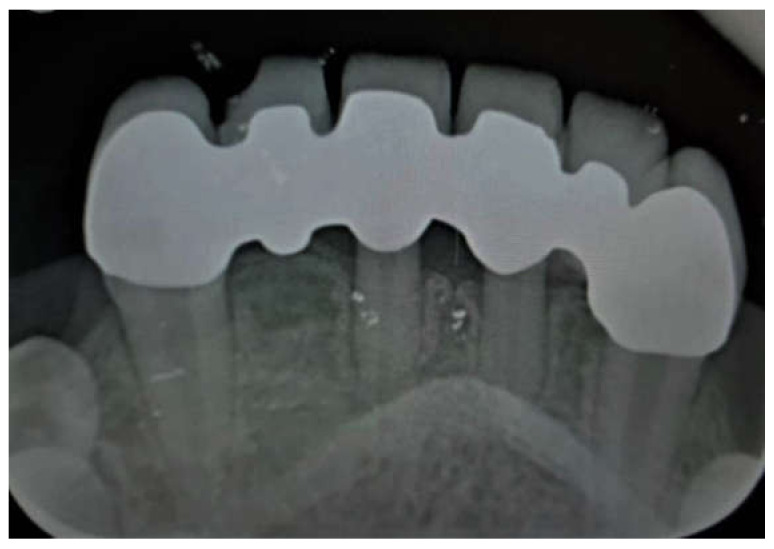
Periapical X-ray. Post-operative, 36 months’ follow-up.

**Figure 6 children-09-00804-f006:**
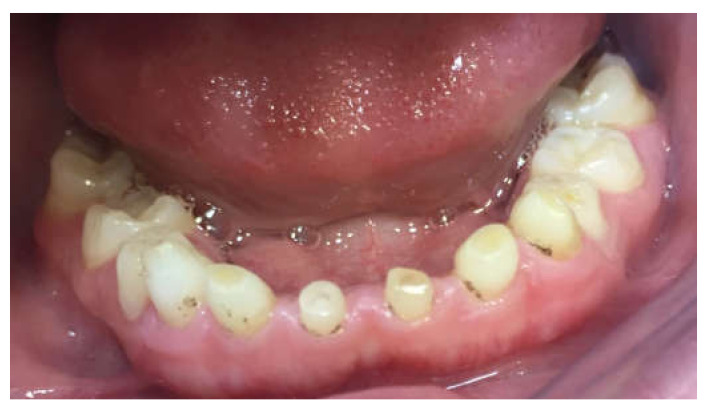
Seventeen-year-old boy presenting with permanent tooth agenesis.

**Figure 7 children-09-00804-f007:**
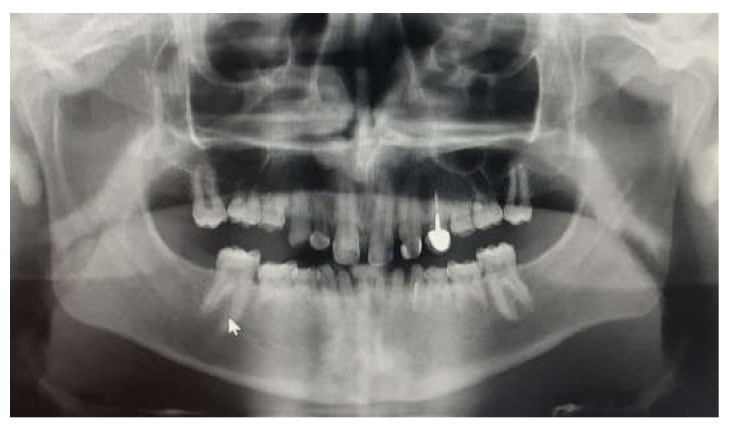
Panoramic pre-operative X-ray.

**Figure 8 children-09-00804-f008:**
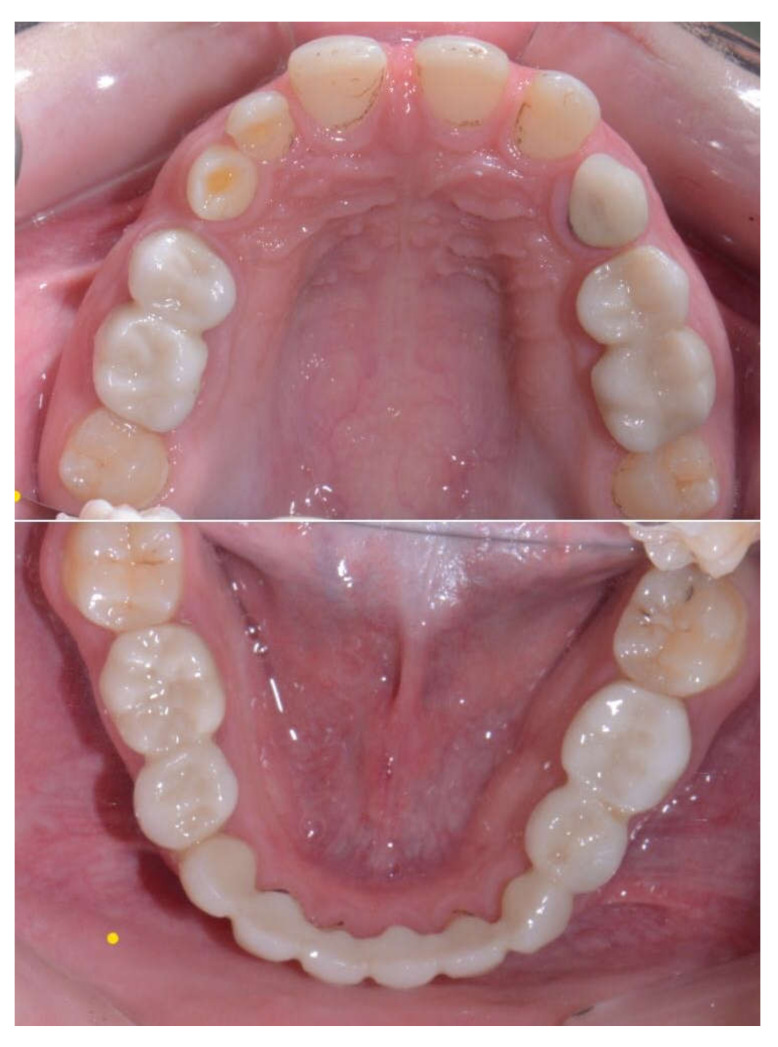
Porcelain teeth restoration 45–35.

**Figure 9 children-09-00804-f009:**
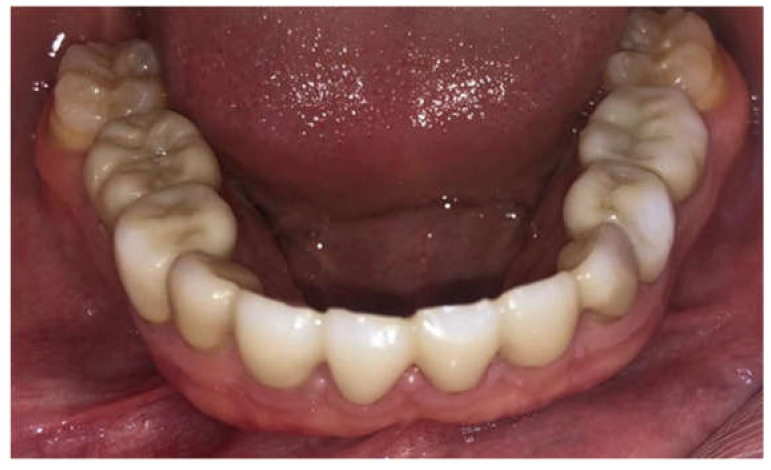
Porcelain teeth restoration 45–35. 60 months’ follow-up.

**Figure 10 children-09-00804-f010:**
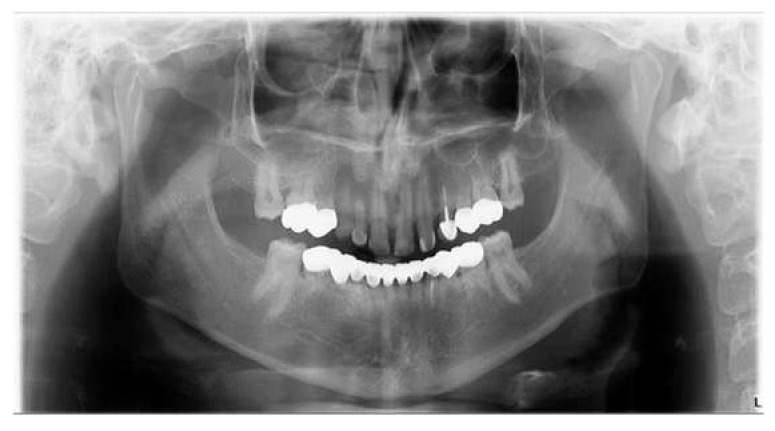
Panoramic X-ray. Post-operative, 60 months’ follow-up.

**Table 1 children-09-00804-t001:** Restorations distribution.

Type	n/N	%
First-Molar	12/58	20.68
Second-Molar	12/58	20.68
Incisor	20/58	34.48
Canine	14/58	24.13

n: tooth type restored; N: total number of restorations.

## Data Availability

The data that support the findings of this study are available from the corresponding author, [S.N.], upon reasonable request.

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
