# Peer review of "Primary Teeth Supported Fixed Prosthesis—A Predictable Treatment Alternative"

_children, 2022, doi:10.3390/children9060804_

Round 1

Reviewer 1 Report

The aim of the retrospective study was to evaluate the clinical and radiological long-term functional and esthetic outcome following restoration of teeth agenesis using primary teeth to support fixed all ceramic prosthesis.

Commonly primary teeth receive composite restoration; hence the approach using is quite new using ceramic material in younger patients.

Some questions occur while reading the manuscript:

  1. Were patients and parents offered any alternative treatment and/or material?
  2. How is the cost-effectiveness of ceramic restorations used in primary teeth?
  3. The abrasiveness of ceramics toward the antagonist teeth should be discussed.
  4. The authors should discuss the alternatives of fixed prosthodontics and also the alternatives for ceramic materials and should present data on survival rate of such treatment options.
  5. This study is very specific in using ceramic material in primary teeth. Due to the small number of included cases the conclusions cannot be generalized.

Author Response

Thanks for the specific comments on different sections of the manuscript corrections  were made.

Keywords
The manuscript presents four keywords. For keywords, where possible, please use Medical Subject Headings terms (MeSH Terms). Initially, none of them is a MeSH term. Alternative MeSH terms proposed could be “anodontia”  better than “tooth agenesis”, or “tooth, deciduous” rather than “primary teeth”. Nevertheless, these suggestions about keywords are optional, not mandatory.

  • The keywords were changed according to the suggestions

General comments
Page 7, line 31. To make text understanding easier, if the author's name appears in the text, place the reference number immediately after the name, not at the end of the sentence or paragraph.
Reference number 36 is not cited in the text when it does appear in the reference list at the end of the manuscript. Please, cite it in the text.

  • Text was corrected

References
Total number of manuscript references: 36.
Nice, the references are adequate and up-to-date. However, according to the journal guidelines, the volume should be indicated in italics, not in plain text. Please, consider presenting the volume number in italics for all references.

Please, adhere to the following general reference format for journal articles:

  1. Author 1, A.B.; Author 2, C.D. Title of the article. Abbreviated Journal Name YearVolume, page range.

  • References were corrected

Figures
Total number of manuscript figures: 9.
In figures 2 and 6 legends, consider replacing the abbreviation “Pre-op” with “Pre-operative”.
In figures 3, 4, and 9 legends, consider replacing the abbreviation “Post-op” with “Post-operative”.

  • Corrections were made

Tables
Total number of manuscript tables: 1.
Please, consider including a table footnote to explain the abbreviations. In the second column, according to the data format presentation (12/58), the header N should be replaced by n/N.  (n: tooth type restored; N: total number of restorations).

  • Corrections were made

Reviewer 2 Report

This retrospective study assesses the long-term functional and esthetic outcomes following the restoration of teeth agenesis using primary teeth to support fixed all-ceramic prosthesis. A clinical and radiological study of fifty-eight porcelain restorations performed on twenty-five patients aged between 10 and 19 years is shown. The manuscript contains four keywords, nine figures, one table, and thirty-six references. Additionally, the study has been approved by a Human Research Ethics Committee. Overall, it is a correct and well-conducted manuscript, although some remarks are made.

Specific comments on different sections of the manuscript are made.

Keywords
The manuscript presents four keywords. For keywords, where possible, please use Medical Subject Headings terms (MeSH Terms). Initially, none of them is a MeSH term. Alternative MeSH terms proposed could be “anodontia”  better than “tooth agenesis”, or “tooth, deciduous” rather than “primary teeth”. Nevertheless, these suggestions about keywords are optional, not mandatory.

General comments
Page 7, line 31. To make text understanding easier, if the author's name appears in the text, place the reference number immediately after the name, not at the end of the sentence or paragraph.
Reference number 36 is not cited in the text when it does appear in the reference list at the end of the manuscript. Please, cite it in the text.

References
Total number of manuscript references: 36.
Nice, the references are adequate and up-to-date. However, according to the journal guidelines, the volume should be indicated in italics, not in plain text. Please, consider presenting the volume number in italics for all references.

Please, adhere to the following general reference format for journal articles:

  1. Author 1, A.B.; Author 2, C.D. Title of the article. Abbreviated Journal Name Year, Volume, page range.

Figures
Total number of manuscript figures: 9.
In figures 2 and 6 legends, consider replacing the abbreviation “Pre-op” with “Pre-operative”.
In figures 3, 4, and 9 legends, consider replacing the abbreviation “Post-op” with “Post-operative”.

Tables
Total number of manuscript tables: 1.
Please, consider including a table footnote to explain the abbreviations. In the second column, according to the data format presentation (12/58), the header N should be replaced by n/N.  (n: tooth type restored; N: total number of restorations).

Author Response

Thanks for the comments corrections   were made accordingly  

  1. Were patients and parents offered any alternative treatment and/or material?
  • Text was added as suggested

“All procedures were fully explained to the patients and parents  alternative treatment and/or material were offered  ,the Ethics Committee of the Tel Aviv University approved the study.”

  1. How is the cost-effectiveness of ceramic restorations used in primary teeth?
  • Text was added in the discussion

 “Commonly primary teeth receive composite restorations (direct or indirect fabrication) the maintenance required over the years (color changing  ,composite fractures) hence the approach of using new ceramic material in younger patients  become popular despite being initially more expensive.

 The last treatment option that was discussed with the family was FPDs, with the advantages of being more retentive and less demanding of the patient. The parents preferred a fixed ceramic prosthesis. FPDs in children become popular due to superior aesthetics and color stability, improved retention and stability.”

  1. The abrasiveness of ceramics toward the antagonist teeth should be discussed.

  • Text and reference were added in the discussion

“The wear of primary teeth hinge on the enamel hardness, enamel and dentin thicknesses, and biting forces in children . Although tooth wear may caused also  by dental materials,a study evaluated the wear of primary tooth enamel against different ceramic (monolithic zirconia, lithium disilicate glass ceramic, resin nanoceramic) and composite resin (nanohybrid resin) materials, indicated that zirconia causes lesser antagonist tooth wear than does lithium disilicate, resin nanoceramic, and nanohybrid composite resin.It may recommended for  used as full coronal coverage in primary tooth restorations.”[25]

  • Bolaca A, Erdogan Y. In Vitro evaluation of the wear of primary tooth enamel against different ceramic and composite resin materials. Niger J Clin Pract. 2019 ,22,313-319.

  1. The authors should discuss the alternatives of fixed prosthodontics and also the alternatives for ceramic materials and should present data on survival rate of such treatment options.

  • Text was added and clarify in the discussion

  1. This study is very specific in using ceramic material in primary teeth. Due to the small number of included cases the conclusions cannot be generalized.
  • Texet was added to the conclusions in order to clarify

“Within the limitations (retrospective, one center) of the present study, further clinical and radiological studies are needed to determine the efficacy of ceramic prosthesis in restoring agenesis teeth using primary teeth as final abutments”

Reviewer 3 Report

The meaning and description of the study is often lost with the poor English translation.

Overall the study has merit, but particularly where long span bridges are used across the arch, longer term results are required. It is difficult to measure growth outcomes for subjects who have these restorations where there is no baseline for actual growth direction and amount. Often the reason for so many denture adjustments when using removable appliances is due to bone growth.

The radiograph needs to include follow-up peri-apical radiographs to assess changes to the tooth roots. 

Author Response

The meaning and description of the study is often lost with the poor English translation.

-----English was revised

Overall the study has merit, but particularly where long span bridges are used across the arch, longer term results are required. It is difficult to measure growth outcomes for subjects who have these restorations where there is no baseline for actual growth direction and amount. Often the reason for so many denture adjustments when using removable appliances is due to bone growth.

---the text was modified as follows "

Fixed prosthesis including rigid connectors are frequently prevented in actively growing patients since they might avert jaw growth, particularly if the prosthesis crosses the midline. Consistent with Barrow and White [26], intercanine width is determined at the ages of 5 - 8 years. Created by distal movement of deciduous canines into the primate spaces to provide space for the erupting permanent incisors. Lack of lower permanent incisors causes canines early eruption leading to little intercanine growth. The principle growth will be distal to the last deciduous tooth to accommodate the eruption of the permanent teeth, by that arch length is increasing. Consequently, only patients ≥10years were included in the present study."

The radiograph needs to include follow-up peri-apical radiographs to assess changes to the tooth roots. 

---Figure 5 was added

Figure 5. Periapical X-Ray. Post-op, 36 months’ follow-up.

Round 2

Reviewer 1 Report

The authors addressed the recommended issues and the paper improved. The paper can be accepted for publication.

Author Response

thanks for your comments